# An Innovative Simulation Agent-Based Model for the Combined Sea-Road Transport as a DSS

Alessandra Renna *, Marco Petrelli [ID], Stefano Carrese and Riccardo Bertocci

Department of Engineering, Roma Tre University, 00146 Rome, Italy; marco.petrelli@uniroma3.it (M.P.); stefano.carrese@uniroma3.it (S.C.); ric.bertocci@stud.uniroma3.it (R.B.)
* Correspondence: alessandra.renna@uniroma3.it

**Abstract:** This research proposes an innovative approach to evaluate modal shift from the road-only to the combined sea-road transport in order to implement new policies and introduce a Decision Support System (DSS) for the transportation planner's decision. The impact of these is carried out by using an innovative simulation tool which has the capability to simulate the real choice process of all stakeholders involved, specifically modelling the freight forwarder's point of view. The model runs as a single-agent based simulation which uses a multimodal network with detailed zoning. The simulation tool, capable of simulating the assignment of the whole network simultaneously, consists of a path choice model and a mode choice model for each o/d pair considered, establishing o/d pairs suitable and not suitable for modal shift. Three policies have been designed and tested through the simulation tool with an application in the Italian context: (1) internalization of the external costs of heavy vehicles; (2) introduction of a bonus for shipping companies; (3) design of new Ro-Ro services. The most affecting policy concerns an increase of speed of some Ro-Ro services to 22 kn, proposing a good balance between the navigation costs and the potential demand attracted.

**Keywords:** modal shift; simulation; policies; Ro-Ro services; intermodal transport

## 1. Introduction

Short sea shipping (SSS) represents an essential aspect of supporting the integration of transport modes, allowing the development of alternative and more sustainable transport solutions. SSS represents one of the main pillars of European Union (EU) transport policies, as it aims to reduce road congestion, shift freight transport from road to short sea shipping and enhance economic and social cohesion between countries. The core of the EU strategy for promoting SSS lies in the Motorways of the Sea (MoS) infrastructure initiative, focused on developing intermodal transport. Roll-on roll-off (Ro-Ro) transport sustains the development of MoS. It offers some advantages for overall transport costs, especially for unaccompanied transport, and an essential reduction of environmental and social costs produced by road transport of freight concerning national context and short shipping distance [1].

Moreover, the geographic topography of some countries influences the introduction and the development of alternative modes of transport instead of traditional road transport. For example, in Italy shipping might be used on many trips, thus entailing the exploitation of this alternative. However, despite the EU's efforts to promote SSS, the combined sea-road transport has not yet reached a significant market share compared to land transport. Looking into the past supporting EU programs for maritime transport, they often seem to offer the establishment of some operational subsidies for the management of maritime routes. For instance, Marco Polo programme provided grants to transport service operators to facilitate a shift from the road to more environmentally friendly modes of transportation, such as SSS or a combination of modes of transport in which road trips are as short as possible (EFTA Bulletin 2007). The lack of satisfactory results of such initiatives suggests the need to adopt another approach, such as developing better models supporting the

definition of policies and new maritime services. Previous experiences in projects and policies definition clearly show the reduced capability of the developed transport models to analyse and forecast the system evolution correctly. Many of these models adopt the point of view of the decision-maker/planner or consider a vast number of agents involving difficulties in simulating their interaction and in carrying out satisfactory results in terms of potentialities and criticalities. A specific complexity characterizes freight transport for the number of actors involved, their relationships and all stakeholders' real needs. Such a system is not described profoundly and thoroughly, and the effectiveness of the defined policies is limited as shown by the impact on modal shift.

At a national level, two incentives were proposed to promote "more sustainable" modes of transport: the Ecobonus (Law No. 265 of 2002) and the Marebonus (Stability Law for the years 2016–2018). In the first case, the incentive was reserved for road transport, based on a minimum number of trips made annually; in the second case, the bonus allocated is paid directly to the shipping companies who freely decide to turn it to the road haulage companies. These experiences and the directives provided at European level highlight the need to focus on the search for alternative and sustainable transport solutions, thus providing the decision-maker with tools capable of grasping the real phenomena in progress, evaluating the goodness of the measures adopted and developing effective implementation interventions concerning the proposed objectives.

Starting from these considerations, the study proposes an innovative approach to evaluate modal shift from the road-only to the combined sea-road transport in order to implement new policies, thus promoting intermodality. The model might stimulate new Ro-Ro services and introducing new policies based on an innovative methodology. These are expected to simulate the real process of modal choice of the stakeholders of freight transport system at the national level, focusing the attention on combined sea-road transport and the road-only alternative. In fact, new policies are introduced and tested through a simulation tool from the outputs of the current scenario and the definition of an analysis model. In this sense, detailed modelling of the maritime and road transport systems permits evaluating the potentiality of the combined sea-road transport, supporting the planner's decision to develop effective freight transport solutions. The methodology has been applied to the Italian context that represents the main route in the intra-EU maritime transport with 97.8 Mln tonnes transported in the year 2018 (EU statistics—DG Move Website).

## 2. Problem Overview

The development of modal alternatives to road transport for the freight is a topic of increasing interest among researchers. Starting from the measures proposed in the White Paper 2001 for the promotion of transport system capable of shifting the balance between modes of transport, Marco Polo programs and the MoS, SSS became one of the critical aspects in developing the Trans-European Transport Network (TEN-T). The attention to SSS derives from its capability to offer competitive and sustainable transport alternatives for road-only transport [2]. It acts by stimulating new services, thus taking advantage by using intermodality and reducing some of the environmental issues related to the emissions from road transport, especially along land transport corridors characterized by high congestion phenomena [3]. However, despite the EU efforts in promoting actions to encourage SSS, during the period of 2000–2009, the modal share of road transport increased by 11.4% whereas shortsea transport grew by only 1.7% [4] and, according to more recent studies [5,6], the results achieved for SSS promotion are very poor. Such information clearly indicates the low impact of the plans promoting SSS over a long period.

The scientific literature, summarized in [7], reports some contributions about SSS, mostly dealing with the assessment of the impacts of new SSS services compared with road-only services in terms of internal and external costs [8–11]. In [8] a methodology is proposed to conduct a quantitative evaluation of the seaside and landside port's accessibility with reference to the intermodal corridor Italy-Spain to identify new potentially attractive maritime services as an alternative to the road-only services. Ref. [9] compares road and sea

freight transport for an internal connection in Greece, aiming to provide a feasible solution towards promoting SSS in the transport chain between mainland ports. Ref. [10] proposes an assessment methodology, based on aggregate discrete choice model, simulating the split between the competitive transport alternatives in the Mediterranean basin. Ref. [11] analyses the environmental performance of alternative modes of freight transport from Trondheim, Mid-Norway, to Paris, France with different degrees of distance travelled by sea and by road to verify if maritime transport solutions are more environment-friendly than road transport alternatives.

Other studies analyse existing services and their development. For example, in [12], service attributes of SSS operations within multimodal transport chains are identified using a questionnaire to logistics operators, shippers' associations and intermodal rail operators. The aim is to estimate the SSS industry, adopting specific service attributes to increase its competitiveness. However, few studies deal with the assessment of specific policies for promoting SSS. One of these proposes a theoretical model to evaluate the implementation of policies based on supporting companies with a project to transfer freight from road to SSS routes [4].

Many other contributions are characterized by very different analyses in terms of approach and structure (quantitative, qualitative, etc.). Many papers describe what happened in a specific context, but these results are not very useful in quantifying how to proceed for a Ro-Ro market increase. In other cases, the analysis makes a useful list of elements to focus on without any quantitative output or analysis.

Freight transport models are still evolving due to the intrinsic complexity caused by the high number of decision-makers, the variety of goods transported, the high variability of decision-making processes and the limited availability of information. In recent years, new models for freight transport simulation have been developed. Over time, these models have seen a progressive evolution by introducing new concepts such as logistical choices, transhipment, storage and procurement of goods. The freight models can be classified into three categories [13,14]: first-generation, second-generation and third-generation.

The first-generation models were the classic 4-stage models (generation-distribution-modal split-allocation), in other words, aggregate-type models that mainly focused on transport. On the other hand, the second-generation models took into account logistical choices and were therefore based on practice, combining elements of micro and macro simulation. Finally, an agent-based or multi-agent-based simulation characterized the third-generation models. In other words, they took into account several subjects involved in the logistics chain and made choices at the company level. Examples of first-generation models are the "Italian National Model System" [15] and the "Transtools" [16]. As a study area, the first one concerns Italy, while the second is a European model. Both represent the classic 4-stage model, but while the Italian one has denser zoning (267 inland areas), the European one has a lower level of zoning (NUTS2). There are also differences concerning the modes of transport taken into consideration: the Italian model focuses on road and rail transport, while the European model also considers maritime transport.

A freight simulation model for comparing sea-road transport and road-only adopting an agent-based modelling approach is presented in [17]. At a national level, the competitiveness between the same transport modes, concerning only routes with Sicily, is evaluated promoting the investment of maritime decision-makers and operators in the improvement of the MoS thus introducing new routes [18].

As a typical example of a second-generation model, the SMILE model can be considered [19]. Its study area is the Netherlands and it has a NUTS2 level zoning (40 internal and 60 external zones) and concerns the modes of transport by road, rail, sea, air transport and the transport of fluids via pipelines. As for the first-generation models, this also starts from the classic 4-stage model (generation-distribution-modal choice-assignment). However, in this case the logistics choices of the transport of goods are also considered, making it a strategic predictive model. The SMILE model is a Decision Support System (DSS) that allows the user to design future scenarios for the transport of goods with a time

horizon of about 25 years. The third generation, as already stated, concerns agent-based or multi-agent-based simulation models, which take into consideration several subjects in the supply chain. For example, the TAPAS model is considered and it is composed of a physical simulator and a decision simulator [20]. The first consists of a graph made up of N nodes and L directed arcs, representing production centres, sorting centres and retailers. In contrast, the decision simulator is made up of six decision-makers: the transport chain coordinator, the product buyer, the transport buyer, the transport planner, the production planner and the customer. The vehicles are also considered in the simulation, characterized by maximum travel speed, load capacity, type of fuel used and emissions. Therefore, the TAPAS model is more effective than traditional approaches regarding the analysis of freight transport, as it models production, demand and all the interactions between the individual subjects of the logistic transport chain.

A simulation type approach characterizes the latest generation models. According to [21], these models strength lies in their ability to reproduce a complex system such as that of the logistic chain in transporting goods. The what-if approach is based on creating a simulation model that aims to reproduce a real phenomenon, in this case, the choice of alternative modes of transport, to make a prediction whose time horizon can be more or less wide.

According to the innovation in modelling highlighted previously, the study's objective is to develop suitable policies in order to promote sea-road transport, updating a previously proposed model [22]. The model developed in the present paper is a single-agent based model, identifiable as a third-generation one, for the modal choice simulation between road-only and the combined sea-road freight transport mode. The model focuses on transport carrier as the single agent in the logistics chain, among the many existing stakeholders, that makes decisions about the transport activity, knowing the needs of goods owner and characteristics of transport alternatives. This is supported by a panel of 400 manufacturing companies' survey results about an overview of their logistical choices [23]. These results show that transport choices are mainly reserved to transport carrier so that about 1/3 of companies does not know the transport mode used by the carrier. Additionally, the update of the previous model is related to more dense zoning from the regional to the provincial level (NUTS3) and by a revision of the reference databases. Therefore, the simulation model developed can quantify the potential demand for combined sea-road transport based on a detailed estimation of costs and times of travel as a combination of various factors such as new Ro-Ro maritime services or subsidies to road haulage companies or shipping companies. Furthermore, its flexible structure, organised in modules, permits introducing actions to promote alternative transport modes to land transport. In fact, ref. [21] demonstrates the effectiveness of specific subsidies involving a considerable increase in the potential demand for the combined sea-road transport implementing suitable policies. Moreover, its application to a Mediterranean corridor [24] shows an increase in the market share of around 25–30% with policy interventions of financial subsidies.

In general, the policies aim to decongest the road network and reduce road externalities (accidents, pollution, emissions). In this sense, previous experiences such as the Ecobonus and current Marebonus represent a good practice to consider for the definition of effective policies. The Freight Leaders Council working group highlights the importance of intermodality in the transport economy, involving time and cost savings but also a decrease in externalities [25]. The Ministry of infrastructure and transport underlines the importance of intermodal freight transport in the national transport economy, working to move towards combined transport through the use of policies, the development of its ports and an enhancement of the infrastructural network [26].

More details regarding existing policies and researches about these are reported in Section 3.

## 3. Methods

### 3.1. Definition of Policies

In the past years many policies have been implemented or proposed in order to promote more sustainable modes of transport. However, the lack of satisfactory results involves more significant efforts to design effective policies. Therefore, it is crucial to consider the results of the previous experiences and the peculiarity of the investigated system and the developed model. The specific tool used for this study can simulate the whole network simultaneously and establish the basin of O/D pairs potentially interesting for implementing new policies. Moreover, the tool is able also to evaluate the impacts of the proposed policies on the system.

A method for defining the policies has been developed in Figure 1, also reporting the policies proposed. The first part of this analysis involves an important phase of in-depth literature review of more than 20 papers and documents. The aim is to build a database of policies with the main measures, actions or proposals reported gathering the better experience or the promising proposal. Different contexts, in terms of mode of transport or in terms of other countries, are considered to evaluate the policy's potentiality to be implemented in this system.

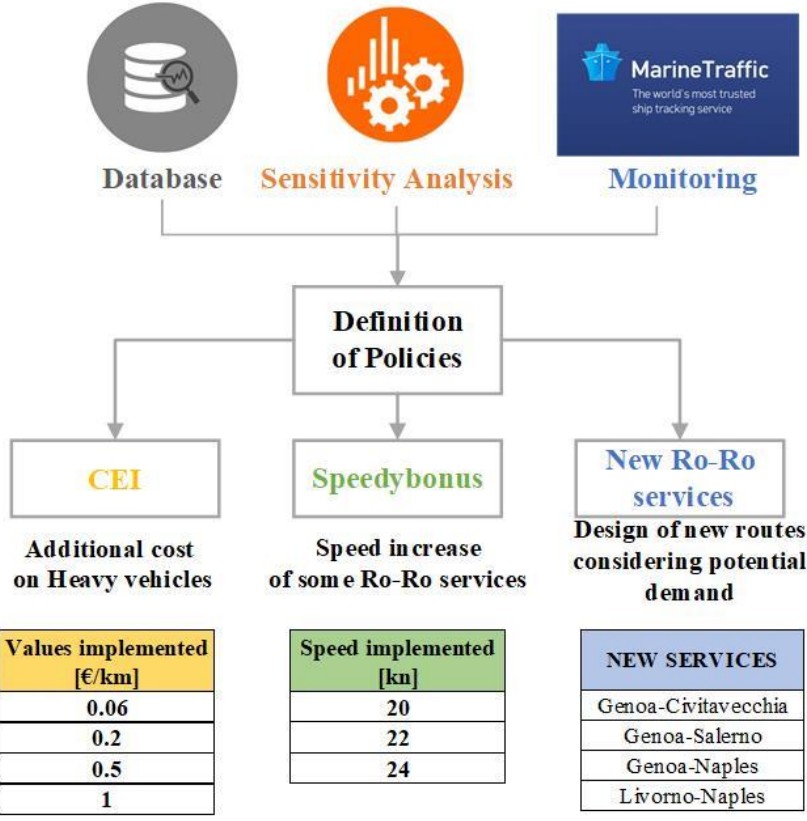

**Figure 1.** Procedure developed for the definition of the policies.

The policies proposed are organized into three types:

- Port management: this macro-category includes all the policies inherent to the port and its surroundings, to ensure its development in terms of operations and services that are carried out within it (speed and efficiency of execution);
- Economic incentives: this category captures all the economic value policies in terms of contributions and incentives for developing combined transport, such as the Marebonus and the Ecobonus experiences. Therefore, all those policies provide economic support through subsidies for transport operators;
- Network and services design: finally, this last category entails policies on network, services and vehicle features to improve the performance of some specific transport el-

ements, such as modifications of characteristics on vehicles (speed increase, frequency or different vehicle fleet) or infrastructure (number of lane or platform conditions) and establishment of new services.

The identification of the most suitable policies is obtained through the sensitivity analysis carried out with the outputs of the current scenario. Due to the lack of information regarding operational management in each port considered, the proposals measures refer only to "Economic incentives" and "Network and service redesign".

As a result, three policies have been designed: (1) Internalization of the external costs of heavy vehicles (CEI); (2) Introduction of a bonus for shipping companies called Speedybonus; (3) Design of new Ro-Ro services.

The first is part of the macro-category of "Economic incentives", while the last is about "Network and service redesign".

The CEI is addressed to all heavy vehicles circulating on the road, regardless of whether they used the road-only mode or the sea-road transport. The concept is to try to internalize the high external costs produced by these vehicles (emissions, accidents, congestion, greater burden on infrastructures), adopting an additional cost (in €/Km) for heavy vehicles. Thus, this policy aims to promote the Ro-Ro services and reduce the externalities produced by road vehicles. The values of this additional cost are fixed, based on the report of Confcommercio of 2018 [27], which contains some reflections on the transport system in Italy and the measures adopted by the neighbour countries (Switzerland and Austria) to reduce pollutants in the Alpine passes. In the first case, a tax (Heavy vehicle fee HVF) has been adopted since 2001 on heavy vehicles with a total gross weight exceeding 3.5 tons, depending on the distance travelled, the gross weight and the vehicle's emission standards while, in the second case, the tax (LKW maut, as called in Austria), introduced since 2004 is associated with the travelled distance in specific natural context.

The Speedybonus considers the relevant influence of shipping times for the competitiveness of combined transport so entailing a contribution for the shipping companies to increase the navigation speed. A speed increase implies a growth in the transport costs due to the relevant increase in fuel consumption. The subsidy permits a speed increase covering the additional costs induced, without changes in maritime rates for road hauliers. Speed is increased for some specific Ro-Ro services with a high demand and a fair number of trips/days obtained. Specifically, the policy is evaluated for the services with a navigation time greater than 9 h testing three reasonable values of speed (20, 22, 24 knots) for reducing travel times.

The last policy "Design of new Ro-Ro services" aims to implement new services, decrease the access/egress distances and consider the origin/destination pairs (O/Ds) potentially interested in modal shift. This policy runs using the simulation tool. In fact, the outputs of the current scenario, highlighting the criticalities, address and support the action of a policy maker. The potentiality for any possible new Ro-Ro service is evaluated by computing the potential basin of each port, considering a road distance of 300 km and a maximum travel times of 4.5 h.

The last part of the policies study focuses on evaluating the policy's impact considering different variables to test the validity and the feasibility of each policy, identifying the most suitable, as shown in the sixth Section. Simple indicators, reported in Figure 2, are introduced to better evaluate the proposed measures: the number of O/D pairs potentially attracted and the related demand (in tons) are computed in all applied policies; CEI entails the computation of the total distances travelled for each transport mode; the Speedybonus requires the computation of the navigation costs concerning fuel consumption due to the speed increase. The potential attraction is considered if the condition on the travel times and on travel costs are both satisfied and the combined alternative is more convenient due to the implementation of the specific policy.

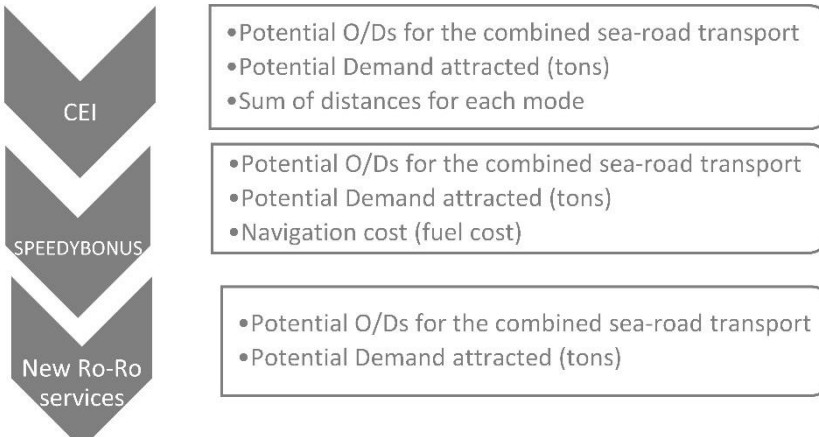

**Figure 2.** Indicators used for the evaluation of the impacts of the implemented policies.

The definition and the evaluation of the policies' impact is carried out by using an innovative simulation model, adopting a new approach capable of considering and better modelling the complexity due to the number of actors involved, their relationships and the real needs of all stakeholders.

### 3.2. Methodology Proposed for the Innovative Simulation Tool

The methodology developed can be considered as a simulation procedure and it is an evolution of the methodology reported in [22]. It is based on the simulation of the road-only and the combined sea-road transport for freight to provide a valid and powerful tool that permits gathering and simulating the freight transport choices concerning stakeholders' needs. The model is implemented for the accompanied transport that represents about half of the traffic in the national network of Ro-Ro routes.

Figure 3 shows the general overview of the methodology, considering the overall flow of data and information and the logical links between each part of the procedure, structured in independent and flexible modules.

Firstly, the input demand requires a procedure to adapt it to the specific zoning system and elaborate any relevant information. At the same time, the network is settled concerning the considered transport modes and adding performance characteristics. Therefore, the methodology acts involving the construction of a specific database capable of integrating the transport modes information and providing the appropriate input data for the following part of the model.

The simulation tool, developed in Python 3.8, consisting of a path choice model and a mode choice model focused on transport operators' point of view, entails the identification of the O/D pairs suitable or not suitable for modal shift. The structure of the tool can simulate the assignment of the whole network simultaneously, considering implicitly the various needs of the other players of the system using specific constraints and time and costs' travel components.

The last part of the methodology consists of an analysis model. It quantifies the potentiality and the criticalities of alternative transport modes concerning the traditional ones. The evaluation of each scenario is also carried out by a sensitivity analysis on the significant variables such as fuel costs, navigation speed, road speed, etc. The outputs of the simulation model provide an overall quantification of the potential modal shift. The tool, recording each travel components, entails a well-structured and accurate analysis of specific performance indicators to highlight the critical points in terms of travel times and costs. Consequently, each modification might be simply implemented on the network model, building different network configuration, or modifying some parameters on the mode choice model of the simulation tool. Therefore, this approach represents a starting point to develop a DSS for freight transport analysis.

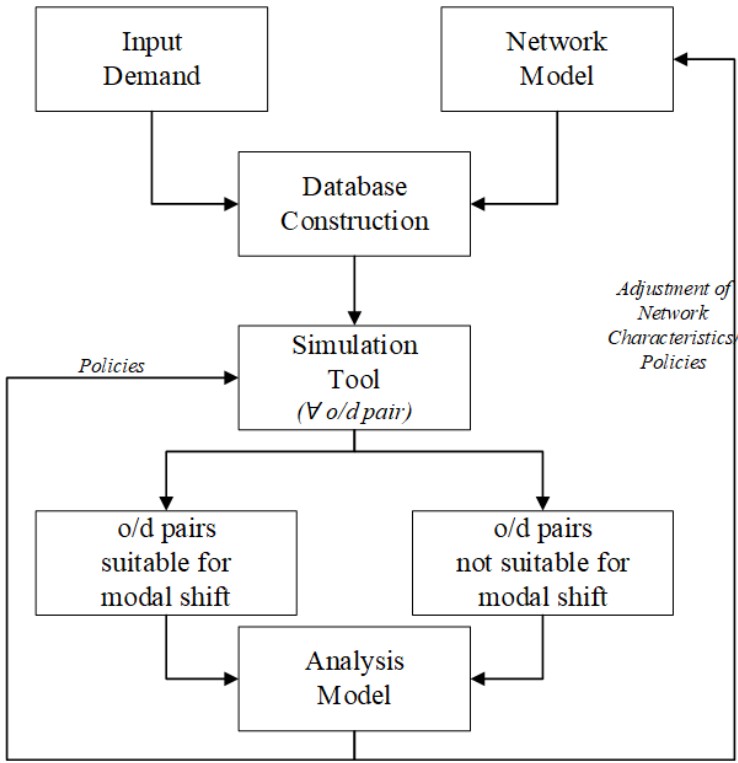

**Figure 3.** General overview of the methodology.

In detail, Figure 4 reports an overview of the simulation tool developing the combined sea-road transport procedure as the alternative transport mode. The structure of the model makes it suitable for other combined transport systems through specific network and demand implementations. The simulation tool is structured in two main phases. First, a preliminary selection phase is carried out to identify O/D pairs potentially interesting for the modal shift; second, an assignment phase defines the potential O/D pairs, solving a path choice problem firstly for each transport mode and secondly for a mode choice model for each O/D pair.

The first phase is made up of three constraints, previously defined in [22] which provide the potential O/D pairs and the relative demand for the combined sea-road transport, starting from an initial freight demand:

- Geographical constraint: it considers the geographical position of origin and destination of the trip concerning alternative transport modes. It permits to remove specific O/D pairs whose trips are necessarily carried out by road-only such as interzonal trips between some regions;

- minimum trip length: considering the specifications and suggestions in [28], an O/D pair can be potentially shifted from road-only transport to the combined sea-road transport if the trip length, computed as the shortest path between origin and destination on the road network, is higher than the threshold value of 300 km representing the minimum value for the convenience of combined transport;

- accessibility criteria: it considers the geographical position of origin and destination of the trip concerning the seaports' location. The match between seaports and hinterland region has been done according to the shortest path, as well as considering existing Ro-Ro services. The O/D pair is selected if the minimum length of the road-only trip is larger than the sum of the access and egress distance to/from the starting/ending port considering the current/new Ro-Ro services.

This preliminary phase consists of a preparation matrix for the second phase. The simulation model begins, building different scenarios according to different supply configurations (depending on considered ports and Ro-Ro services with specific characteristics

such as length, travel times, speed, etc.). It requires the construction of a multimodal network consisting of road and maritime links characterised by volume-delay function (VDF's) and nodes (centroids or road and port nodes) with associate penalties due to congestion delays or loss of time in correspondence of operational activities such as loading/unloading operations and ticket control at ports. Then, the Ro-Ro services are implemented introducing their effective length, speed, the type of ships (capacity and technical characteristics), the actual number and network connection. In addition, the network entails intermodal links characterised by a lower speed in order to move from to one mode to another.

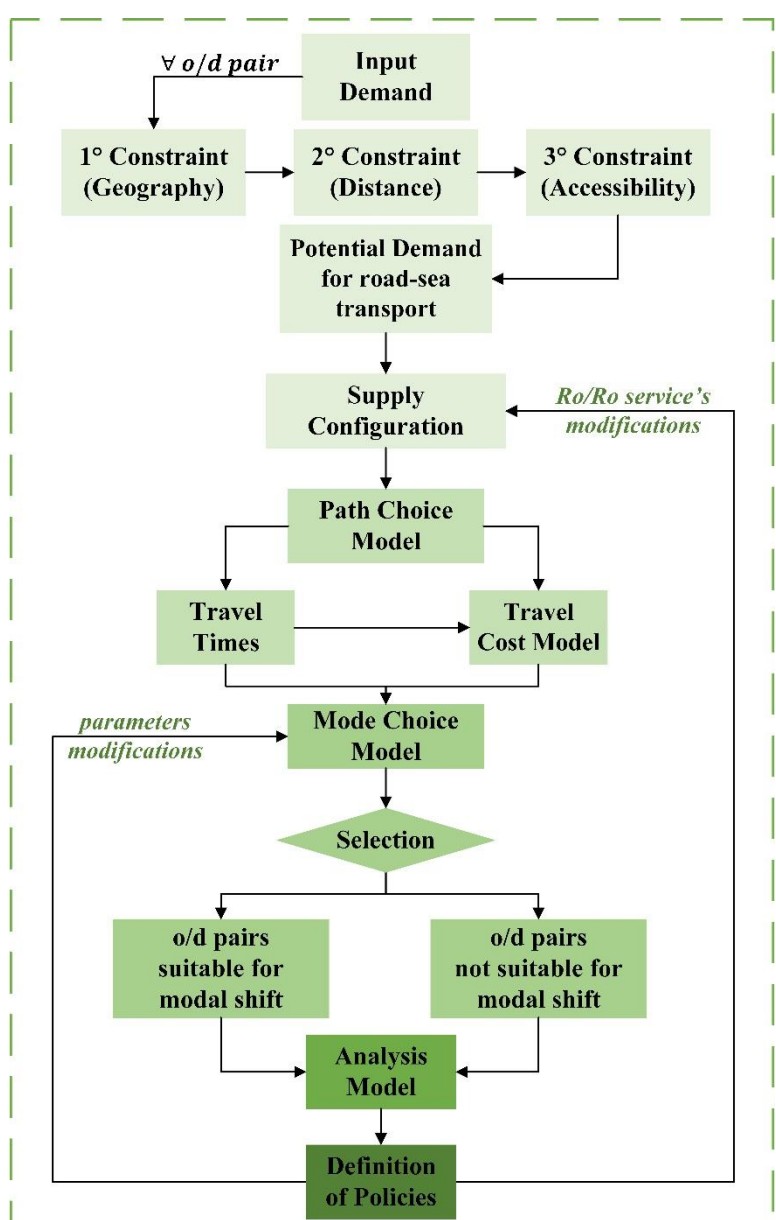

**Figure 4.** Overview of the simulation tool.

For road-only and sea-road transport, the shortest path is defined concerning travel times of the road haulier, considering several time components such as driving times and shipping times, drivers' rest time and loading and unloading times in port. Then, a travel cost model is implemented to consider the cost components that depend on travel times and effective road distances, assuming the model is developed as accompanied transport. This model was developed in [22] referring to "Indicative reference values of the operating costs of the road haulage company on behalf of third parties" (Article 1, paragraph 250 of

the Law of 23 December 2014, No. 190—Law 2015 stability). Such a model is composed of truck, insurance, fuel, salaries, maintenance, motorway tolls and company cost. In addition, the salaries depend on drivers' working time, considering the average wage, including transfers and overtime.

The combined sea-road transport entails the specifications of the maritime costs, computed as shipping rate corresponding to the average value paid by the road haulage company to travel on the ship. Data are obtained by the simulation of trips on freight booking platforms concerning existing domestic Ro-Ro services. However, due to the usual difficulties collecting such data, a clustering algorithm (k-means), which associates the rate taking into account the ship's capacity and the connection's length, provides the whole definition of Ro-Ro service's rate.

The route choice model, based on all-or-nothing assignment techniques, provides the minimum paths for each mode. The structure of the simulation tool provides two sets of solutions, distinguishing travel times (T*) and travel costs (C*) minimization, as the alternative to be compared to road-only transport (Figure 5). The difference is based on a reasonable assumption considering that different needs are required involving different type of commodities. For example, the route choice for perishable commodities transport involves identifying the minimum travel times route and, in a second step, selecting the alternative providing the minimum transport costs. In many other cases, the route choice is driven by the minimization of the travel costs and, in a second step, by selecting the alternative also characterized by the minimum travel times. This specification permits the simulation of different needs in path choice for freight transport.

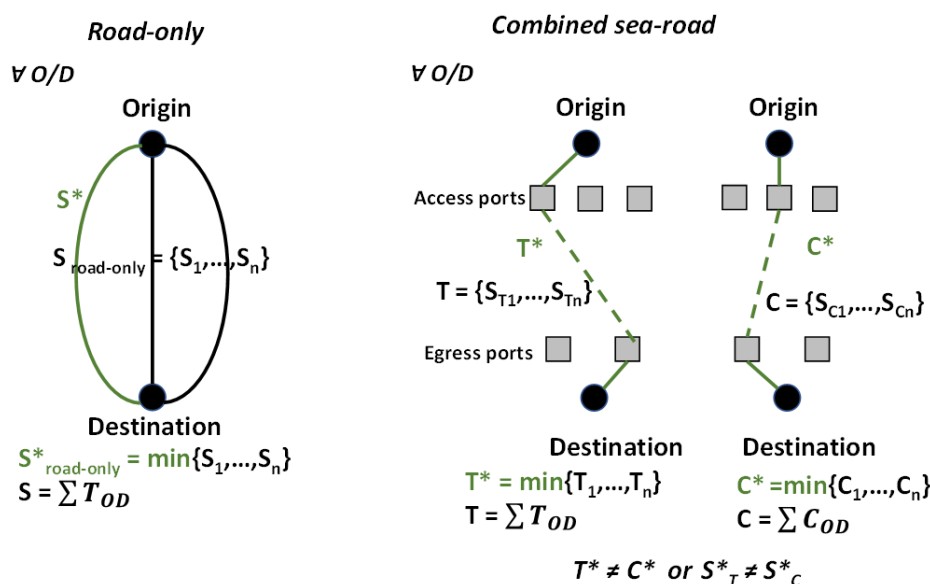

**Figure 5.** General scheme of path choice model for each transport mode.

Specific characteristics of the O/D connection and the type of commodities influence the choice of the most relevant variable and, consequently, the approach might change. In general, the transport costs are the first decision variable for the freight since it represents the goal of choosing a transport mode. The outputs are then used as input for applying the mode choice model, which compares travel times and travel costs for each O/D pair concerning the alternative transport modes. The O/D pairs verifying both the condition based on the travel times and the ones on travel costs are finally selected as potential demand for the combined sea-road transport (O/D suitable for the modal shift). The comparison is based on a sequential phase analysis starting from travel times or travel

costs according to the specific characteristics of the O/D connection. Thus, the travel times and travel costs represent the key variables considered in freight transport.

$$\begin{cases} T_{combined} < T_{road-only} + \alpha \; \forall \, O/D \\ C_{combined} < C_{road-only} + \beta \; \forall \, O/D \\ \alpha, \beta = \text{thresholds values for different Scenarios} \end{cases} \tag{1}$$

The path and the mode choices are focused on finding the more convenient transport mode while the travel time might be a constraint in terms, for example, of the temporal delivery window.

For the comparison between alternatives modes, different thresholds values ($\alpha$, $\beta$) are fixed because the convenience in modal shift could not depend on the strict adherence to a comparison result. However, there is the need to consider a more comprehensive range of acceptability values of the modal shift as really happened in the real world.

The simulation tool identifies the critical points of the investigated system, distinguishing the O/D pairs suitable and not suitable for modal shift. Furthermore, it provides many outputs for each O/D pair (access and egress time, route chosen, speed of Ro-Ro services, total travel times and travel costs, road distance, etc.) due to the mode choice model. This information is helpful to quantify each component considered in the model representing the starting point of the following phase, under development, which allows understanding how to improve the Ro-Ro service network and increase the number of O/D pairs suitable for modal shift. In this sense, it seems crucial to define indicators suitable for the specific model developed. The importance of indicators stands both in the sensitivity analysis referred to each variable and the evaluation of scenarios that implement policies or network modifications. Furthermore, considering that transport simulation models are commonly used to address policymakers' decisions with the aim to promote alternative transport modes, introducing specific and clear indicators provides an effective tool able to evaluate the goodness of project proposals.

## 4. Results and Discussion

This section reports a description of the case study and the network model developed. The results of the current scenario allow the identification of the criticalities. Moreover, the policies proposed in Section 4 are implemented, highlighting the effects on the potential demand for the combined sea-road transport, with the computation of indicators able to gather their impact on the investigated system.

### 4.1. Case Study

The methodology has been applied to the Italian national road and maritime network, considering the existing maritime Ro-Ro services and the elements of the national road network to assure the connections between zones and the access/egress to/from the ports. By referring to the current maritime services, all the seaports currently used for Ro-Ro traffic are considered in the study. Intermodal links represent the connectivity between the road network and the maritime services, considering a lower speed due to both the congestion at ports and loading/unloading operations. The multimodal network includes the nodes representing 27 ports and a zoning system with 107 traffic zones following the NUTS3 regional classification.

Information required by the supply model, such as trip lengths and travel times by road or maritime services and shipping rates, are collected by different open-sources databases (port authorities, maritime companies, national authorities, web-based open data). The freight demand data related to the road-only mode derives from the Italian Ministry of Infrastructure and Transport website, adopting the zoning of the system previously mentioned. The resulting multimodal network is reported in Figure 6.

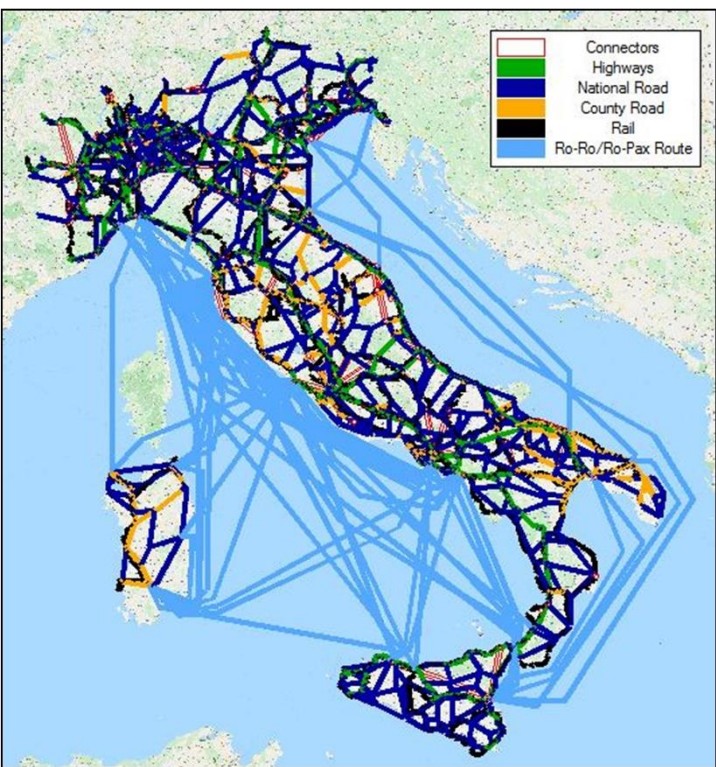

**Figure 6.** Road and maritime Italian network with zoning system NUTS3.

The first numerical application concerns 53 existing Ro-Ro/Ro-Pax services existing in 2019, gathered with a monitoring operation of Marine Traffic open source: 24 Ro-pax services, 18 multi-ports services and most Ro-Ro services are round trip. The routes are mainly located in the Tyrrhenian Sea, as the density of ports for Ro-Ro traffics, while only three services are operating in the Adriatic Sea. Only four routes are North-South connections. The others, in some cases composed by more calls, permits to connect Sicily and Sardinia with the Italian peninsula. Thus, the maritime network shows both a higher degree of fragmentation and a real need to increase the connections available in order to improve modal shift.

The Ro-Ro rates are carried out by k-means clustering algorithm. About 50% of shipping rates are collected through shipping companies' price lists and websites (cargo section) and booking simulation using standard characteristics. The preliminary correlation and regression analyses permit identifying the relevant and independent variables (distance, speed, frequency and capacity in lane meters) to be used in the algorithm to assess the Ro-Ro rates. The results show two different sets based on capacity in lane meters (LC). The first one (LC > 2500) is related to rates increasing with the distance while the second one (LC < 2500) is about rates not correlated with the distance of the route.

The model outputs show that about 20% of O/D pairs are suitable for modal shift while about 40% of O/D pairs do not find a real convenience in the modal split. Therefore, a detailed analysis is possible at the level of every O/D pair. For instance, two solutions are obtained for a specific O/D pair (Torino-Avellino), starting from travel times minimization and travel costs minimization (Figure 7). This last case involves the longest Ro-Ro services while travel times minimization involves the shortest ones.

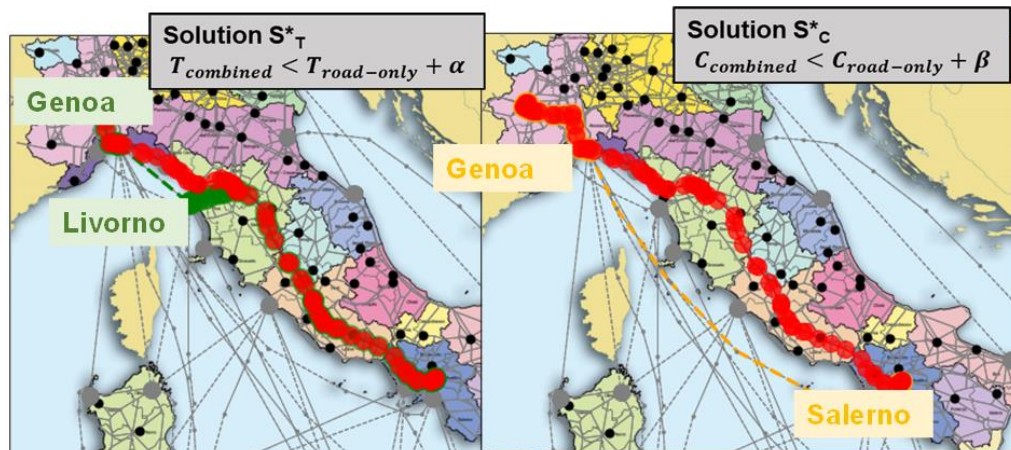

**Figure 7.** Example of Path Choice model for a specific O/D pair differentiating between travel times minimization and travel costs minimization.

Shipping times, representing the most affecting variable for the travel time, indicates the need to analyse the speed of existing Ro-Ro services, introducing more direct services and eventually designing a different maritime network. Finally, the basin of influence for each port represents another output of the simulation tool that is useful for implementing new policies. For example, considering the potentiality of each port in terms of Ro-Ro services, the implementation of new routes might increase the number of O/D pairs gathered by ports.

### 4.2. The Effects of the Policies and Their Impacts

The effectiveness of the policies implemented is reported in terms of the number O/D pairs and tons potentially attracted by the sea-road transport distinguishing travel times (Solution T*) and travel costs (Solution C*) minimization. Moreover, each policy is evaluated by introducing indicators in order to establish their impact and their feasibility. Figure 8 shows the number of the O/D pairs and the tons potentially attracted for each policy implemented and computed without any policy application for the current scenario. It should be considered that the results are expressed considering a threshold value ($\alpha$ and $\beta$) equal to 0. Therefore, by adding reasonable threshold values, the convenience in modal shift could not depend on the strict adherence to a comparison result, as really happened in the real world, and the O/D pairs for the combined transport might increase. The CEI brings important benefits for the Solution T*, higher as the additional cost increases. Concerning Solution C*, it is possible to register a minimum improvement for 0.06 €/Km and then a worsening for the other values. This happens because the set of O/D pairs attracted has already reached the best solution for travel costs minimization, so higher values of CEI have no impact. The same trend is obtained for the tons potentially interesting for modal shift.

The Speedybonus entails improvements for both solutions (especially for that one which minimizes costs). Shipping time is the most affecting variable, which significantly impacts the competitiveness of the sea-road alternative. Therefore, the improvements increase as the speed value increases, reducing the time gap between the two modes of transport.

Two new Ro-Ro services (Genoa-Salerno and the Genoa-Naples) lead to considerable improvements. This result suggests that a more performing service for the North-South route could attract demand promoting the use of the combined sea-road transport.

For Solution T*, almost all the policies involve increasing convenient O/D pairs (except the new Livorno-Naples and Genoa-Civitavecchia services). The most affecting policies are the CEI = 1 and the speed of navigation at 24 kn. However, the other values of CEI and speed bring important results, especially as the additional costs or the speed increases. Among the new services, only the Genoa Naples and the Genoa/Salerno have interesting effects. In the Solution C*, the CEI has a modest impact. The values of 0.20/0.50/1 €/Km

involve only slight worsening. Differently, Speedybonus has significantly higher positive effects compared to the Solution T*. Finally, the introduction of new Ro-Ro services has lower effects.

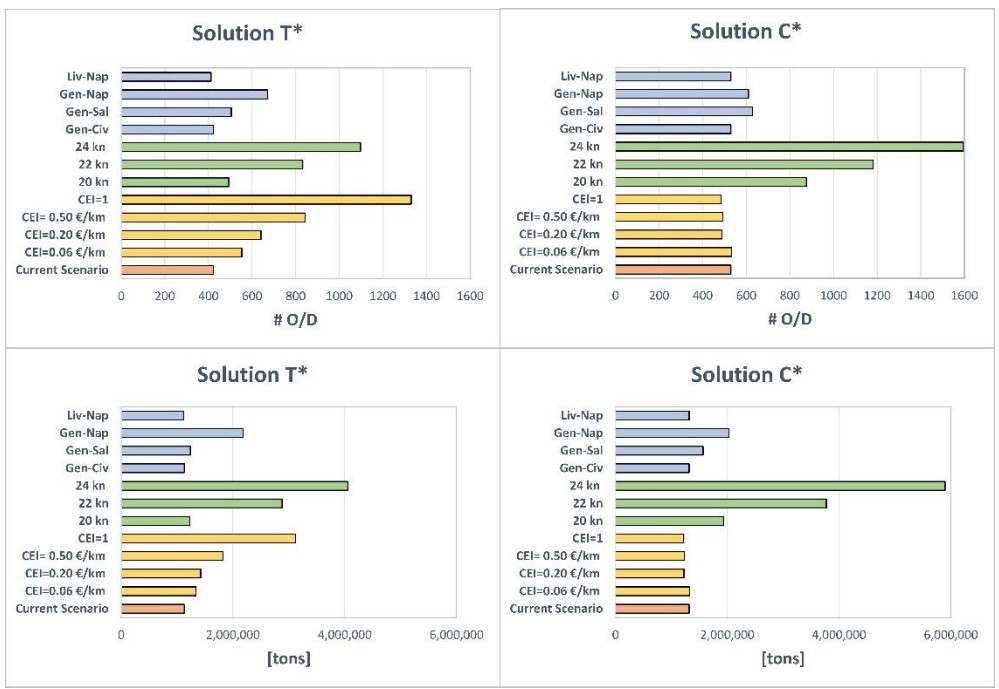

**Figure 8.** O/D pairs potentially interesting for the sea-road transport e the tons potentially attracted for these O/D pairs implementing the policies.

For Solution T*, an increase in attracted demand occurs in scenarios with the CEI equal to 1 €/km, speeds of 22 and 24 kn, and the introduction of Genoa-Naples Ro-Ro services. For the demand (tons attracted), a speed of 24 kn represents the best scenario, while for the O/D pairs, the more significant effects derive from the CEI equal to 1 €/Km. Finally, the demand potentially attracted by new Ro-Ro services is lower than the number of O/D pairs suitable for modal shift.

For the Solution C*, the demand analysis mainly follows the O/D pairs, except for the Genoa-Salerno and Genoa-Naples services. The CEI policy leads to fewer tons handled while the Speedybonus involves a general increase, especially for 22 and 24 knots (maximum increase). As a result, Genoa-Salerno and Genoa-Naples registers a slight increase in tons moved while the other Ro-Ro services handle the same tons of the current scenario.

Evaluating the impacts of the CEI and the Speedybonus requires the introduction of other suitable indicators that support the feasibility of the proposed measures in the real world. The CEI implies a variation of the travelled distances, as shown in Table 1, with a consequent increase in travel costs. For Solution T*, the road-only distances decrease as the CEI rate rises. Furthermore, for the sea-road transport an increase of access and egress distances and also of maritime ones occurs because O/D pairs are led to choose the combined alternative. For example, for CEI = 1 €/km, the road-only decreases of about 45% while the combined has a greater than 200% growth. Otherwise, for Solution C*, there is an increase in the road-only as the additional cost increases and a decrease in total combined distances. In this specific solution, this measure has no positive effects on the number of O/D pairs suitable for modal shift. Specifically, for CEI = 1 €/km the road-only increase of about 18% while the distances of access and egress distances decrease by about 10%.

**Table 1.** Evaluation of CEI considering the variation of travelled distances for each scenario.

| | Solution T* | | | | Solution C* | | | |
|---|---|---|---|---|---|---|---|---|
| *Scenario* | Total Distance Road-Only (Km) | Total Distance Sea-Road (Km) | Access + Egress Distance (Km) | Maritime Distance (Km) | Total Distance of Road-Only (Km) | Total Distance Sea-Road (Km) | Access + Egress Distance (Km) | Maritime Distance (Km) |
| *Current scenario* | 2,243,953 | 409,963 | 269,822 | 140,141 | 2,508,004 | 547,728 | 211,368 | 336,360 |
| *CEI = 0.06 €/Km* | 2,145,171 | 536,852 | 353,391 | 183,461 | 2,629,848 | 548,960 | 211,905 | 337,055 |
| **Δ (%)** | **−4.40** | **30.95** | **30.97** | **30.91** | **4.86** | **0.22** | **0.25** | **0.21** |
| *CEI = 0.20 €/Km* | 1,994,351 | 622,330 | 407,181 | 215,149 | 2,859,262 | 516,140 | 194,650 | 321,490 |
| **Δ (%)** | **−11.12** | **51.80** | **50.91** | **53.52** | **14.01** | **−5.77** | **−7.91** | **−4.42** |
| *CEI = 0.50 €/Km* | 1,665,126 | 849,559 | 534,224 | 315,335 | 2,974,062 | 516,352 | 195,757 | 320,595 |
| **Δ (%)** | **−25.79** | **107.23** | **97.99** | **125.01** | **18.58** | **−5.73** | **−7.39** | **−4.69** |
| *CEI = 1 €/Km* | 1,229,213 | 1,337,711 | 851,246 | 486,465 | 2,980,370 | 506,161 | 188,993 | 317,169 |
| **Δ (%)** | **−45.22** | **226.30** | **215.48** | **247.12** | **18.83** | **−7.59** | **−10.59** | **−5.71** |

A speed increase involves a growth in navigation costs (see Table 2) for the shipping companies due to the bunker fuel consumption. Therefore, it seems essential to relate the thresholds of speed to the variation of the demand potentially attracted (tons). In fact, an increase in speed makes the services more attractive. Solution C* entails higher values of demand than Solution T*. Regarding the demand, the range varies between 100 k tons and 70 O/D pairs attracted (Solution C* and 20 kn) and over 4.5 million of tons and 1064 O/D pairs attracted (Solution T* and 24 kn). The speed at 22 kn represents the threshold that involves a more significant impact in terms of demand.

**Table 2.** Evaluation of Speedybonus considering the additional navigation costs and additional demand.

| | Solution T* | | | Solution C* | | |
|---|---|---|---|---|---|---|
| **Scenario Speedybonus** | **Additional Cost [€]** | **# O/D Pairs** | **Additional Demand [tons]** | **Additional Cost [€]** | **# O/D Pairs** | **Additional Demand [tons]** |
| 20 kn | 5,769,251 | 70 | 98,169 | 14,333,189 | 346 | 612,560 |
| 22 kn | 31,892,670 | 409 | 1,749,393 | 48,588,214 | 651 | 2,454,455 |
| 24 kn | 69,008,277 | 674 | 2,923,491 | 123,766,910 | 1064 | 4,578,435 |

Therefore, the fuel costs are computed concerning speed variation, comparing the obtained costs with the incentives of the Ecobonus and the Marebonus, as shown in Table 3. Comparing the hypothetical expenses deriving from the policy proposal with the various incentives issued in recent years, the data clearly show that they are comparable with the exception of the higher value observed, related to the 24 kn solution. According to this comparison, it is clear that this policy might be feasible in the real world.

**Table 3.** Value of the incentives in Italy (2007–2026).

| | Ecobonus | | | | Marebonus | | | |
|---|---|---|---|---|---|---|---|---|
| **Period** | **2007–2010** | **2016** | **2017** | **2018** | **2019** | **2020** | **2021** | **2022–2026** |
| Value [Mln €] | 240 | 45.4 | 44.1 | 48.9 | 30 | 20 | 25 | 19.5 |
| Source | AdsP Northern Tyrrhenian Sea | | | | MIT | | Stability law 2021 | |

The results and the procedure developed for evaluating the policy's impact demonstrate the effectiveness of a possible application of the proposed measures. In addition, the evaluation of the impacts permits identifying the most effective policies and considering the implementation aspects.

The most affecting policy concerns an increase of speed of some Ro-Ro services to 22 kn, proposing an outstanding balance between the navigation costs and the potential demand attracted. In addition, it seems important to evaluate a redesign of the service

network to add demand on the routes, promote intermodality and better design suitable Ro-Ro services according to the real needs of the demand.

## 5. Conclusions

The analysis of the initiatives promoted by EU for developing combined sea-road transport shows a lack of satisfactory results and suggests the need to adopt better and more effective models supporting the definition of policies and the forecast of the system evolution correctly. Starting from this observation, the simulation model developed and described in this paper is a single-agent based model, identifiable as a third generation one, for the modal choice simulation between road-only and the combined sea-road freight transport mode. The model focuses on transport carrier as the single agent in the logistics chain, among the many stakeholders existing, that makes decisions about the transport activity, knowing the needs of goods owner and characteristics of transport alternative. This model represents a robust tool capable of understanding and reproducing the complexity of the system investigated. Its structure, quite flexible and simple, permits to evaluate the weakness and critical points of the simulated transport modes. The research represents an innovation regarding other models able to work with real-world data and instances, being a starting point for developing a DSS for freight transport.

The model application permits the definition of new policies to improve the performance of existing Ro-Ro services or to identify new routes that better meet the needs of potentially interesting demand. Specifically, three policies have been designed and tested through the simulation tool, showing the effectiveness of a possible application of the proposed measures. The evaluation of the impacts permits identifying the most effective policies and considering the implementation aspects. The most affecting policy concerns an increase of speed of some Ro-Ro services to 22 kn, proposing an outstanding balance between the navigation costs and the potential demand attracted. It also seems important to evaluate a redesign of the service network to add demand on the routes, promote intermodality and better design suitable Ro-Ro services according to the real needs of the demand. This approach with the construction of a simulation model of the whole transport network represents a novelty developing a DSS able to analyse different conditions of supply and demand so supporting the policy-makers activities, also for large scale context.

Even if the model developed is calibrated and tested in the Italian context, the approach proposed can be easily modified and adjusted for studying other transport modes as the combined transport by railways or other regions.

About the refinement of the present study, one of the main critical issues is the need of collecting additional and reliable data as, for instance, about maritime fares. Therefore, an integration of data and information (e.g., using focus groups or another source of information) could be very useful in order to improve the model capacity of describing the transport system for the freight, characterised by complex relationships among actors. In this way, especially by working in focus groups of stakeholders, it is also possible to validate the proposed approach and understand the evaluation of the proposed policies. Another important element for the refinement of the developed simulation model could be the analysis of the policies simulated also by the point of view of the environmental impacts. Such observation derives by the results obtained with the Speedybonus that is effective on promoting modal shift from road to combined transport but, at the same time, might produce a situation in contrast with the decarbonization policy promoted currently by the EU.

**Author Contributions:** Conceptualization, A.R., M.P. and S.C.; methodology, A.R. and M.P.; software, R.B.; validation, A.R.; formal analysis, A.R. and M.P.; investigation, A.R.; resources, M.P.; data curation, A.R. and R.B.; writing—original draft preparation, A.R. and R.B.; writing—review and editing, A.R. and M.P.; visualization, A.R.; supervision, S.C. All authors have read and agreed to the published version of the manuscript.

**Funding:** This research received no external funding.

**Conflicts of Interest:** The authors declare no conflict of interest.

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
