# Peer review of "An Innovative Simulation Agent-Based Model for the Combined Sea-Road Transport as a DSS"

_sustainability, doi:10.3390/su131910773_

Round 1

Reviewer 1 Report

Authors need to decide which spelling they want to use. Either English (UK) or English (USA). Not both at the same time.

Figures must be included in the right place and mentioned in the body text of the paper. So authors are invited to check this.

The solutions T and C must be clearly indicated in the methodology, which is not the case.

In addition, please deal with the remaining comments.

Author Response

Dear reviewer

thank you for your very useful suggestions. As follows you can find the response to any comment provided: 

Authors need to decide which spelling they want to use. Either English (UK) or English (USA). Not both at the same time.

Reply to comment: the comment is noted and the manuscript is changed accordingly using the spelling of the UK English

Figures must be included in the right place and mentioned in the body text of the paper. So authors are invited to check this.

Reply to comment: the comment is noted and the manuscript is changed accordingly

The solutions T and C must be clearly indicated in the methodology, which is not the case.

Reply to comment: the comment is noted and a reference is included in the text of the manuscript

In addition, please deal with the remaining comments.

Reply to comment: the remaining comments are noted and several sentences of the manuscript are changed as well as the other requests of correction (inclusion of acronyms, split of sentences, clarification about sentences and meaning of the text, etc)

Reviewer 2 Report

Dear Authors, following issues should be taken by you into consideration towards quality improvement of your paper:

  1. The title is a bit misleading when regarding the area of research - your model was tested only in the framework of Italy's transport system. Therefore the title should relate to this fact somehow.
  2. 2. In the title, the abbreviation of DSS should be explained very near, i.e. in the abstract. Now it is just on the 4th page. Can be misunderstood with the Deep Sea Shipping.
  3. To the keywords I suggest to add: modal shift, intermodal transport.
  4. The last part of the section 1. Introduction - about the paper's organization - seems to be unnecessary, very seldom found in research papers. Especially, when the structure is somehow enforced by the MDPI.
  5. Regarding systems supporting modal shift and/or transport greening there are more than only two examples self in the EU. You've mentioned only EcoBonus and MareBonus. Cannot see the criteria of selection these ones. Is this because refering to Italy? Than EcoBonus is not the case (Spanish case).
  6. Very poor literature review, especially regarding modal shift, intermodality, SSS and EU transport policy.
  7. Figures should be improved in the technical aspect - lacking of lines, blurred letters (most probably print screens used).
  8. In the conclusions section could be valuable to add few sentences relating elaborated model to the more global scale/use in other conditions/regions, and developed some conditions to be considered if so. Besides, there are some repetitions from introduction giving no added value.
  9. Please unify also names of cities - sometimes one can meet Genova, and afterwards Genoa (i.e. p. 15, row 10 - both versions in one row).

Self to the idea of the research and proposed model and, in the consequence, 3 options considered for the transport policy - it is discussable but fully allowed to all as authorship. Maybe the further relation to the policy-makers is less clear for me.

Author Response

Dear reviewer

thank you for your very useful suggestions. As follows you can find the response to any comment provided: 

Dear Authors, following issues should be taken by you into consideration towards quality improvement of your paper:

  1. The title is a bit misleading when regarding the area of research - your model was tested only in the framework of Italy's transport system. Therefore the title should relate to this fact somehow.

Reply to comment 1: the comment is noted and the specific information about the tests only in the framework of the Italian transport system is inserted in the abstract as well as better explained in the manuscript.

  1. 2. In the title, the abbreviation of DSS should be explained very near, i.e. in the abstract. Now it is just on the 4th page. Can be misunderstood with the Deep Sea Shipping.

Reply to comment 2: the comment is noted and the DSS meaning is explained in the abstract and this specific aspect is reported also in the first section of the paper.

  1. To the keywords I suggest to add: modal shift, intermodal transport.

Reply to comment 3: the comment is noted and the keywords are modified according to this suggestion.

  1. The last part of the section 1. Introduction - about the paper's organization - seems to be unnecessary, very seldom found in research papers. Especially, when the structure is somehow enforced by the MDPI.

Reply to comment 4: the comment is noted and we agree with reviewer’s suggestion. The last part of the introduction is deleted.

  1. Regarding systems supporting modal shift and/or transport greening there are more than only two examples self in the EU. You've mentioned only EcoBonus and MareBonus. Cannot see the criteria of selection these ones. Is this because refering to Italy? Than EcoBonus is not the case (Spanish case).

Reply to comment 5: the choice of mentioning EcoBonus and MareBonus as well as Marco Polo programme is due to the importance of these inititives especially by the Italian point of view.

  1. Very poor literature review, especially regarding modal shift, intermodality, SSS and EU transport policy.

Reply to comment 6: the comment is noted and in the manuscript is included another updated paper about EU policies effectiveness. We agree with reviewer that additonal citations of papers about modal shift, intermodality, SSS and EU transport policy could be included in the literature review but we think that the manuscript is first of all focused on the description of a new simulation model for this specific sector of the freight transport. Taking into account this aspect, the literature review in our manuscript, is organized to a) identify criticalities about policies promoted in the last years for the modal shift from all road transport and b) describe characteristics of existing simulation models useful for testing and identifying new policies more effective.

  1. Figures should be improved in the technical aspect - lacking of lines, blurred letters (most probably print screens used).

Reply to comment 7: the comment is noted and the manuscript is changed accordingly.

  1. In the conclusions section could be valuable to add few sentences relating elaborated model to the more global scale/use in other conditions/regions, and developed some conditions to be considered if so. Besides, there are some repetitions from introduction giving no added value.

Reply to comment 8: the comment is noted and the conclusions section is changed according to the suggestions received.

  1. Please unify also names of cities - sometimes one can meet Genova, and afterwards Genoa (i.e. p. 15, row 10 - both versions in one row).

Reply to comment 9: the comment is noted and the name of the cities is cited in an unique way.

Self to the idea of the research and proposed model and, in the consequence, 3 options considered for the transport policy - it is discussable but fully allowed to all as authorship. Maybe the further relation to the policy-makers is less clear for me.

Round 2

Reviewer 1 Report

As authors check the reviewed file they will find some comments and formatting issues to be dealt with.

Authors need to edit the text. Sentences are too long and difficult to read. The text must be English proofread by a native speaking person. The file attached shows some of the issues found.

Proper use of terminology is needed. For instance, we do not have shipping fares, but shipping rates or freight rates.

Also, the authors suggest a speed increase in some routes. Is this compatible with the current EU decarbonisation policy? please consider and reflect this in the outcome of the paper?

Author Response

Dear reviewer

thank you for your very useful suggestions. As follows you can find the response to any comment provided:

As authors check the reviewed file they will find some comments and formatting issues to be dealt with.

Reply to comment: the comments included in the reviewed file are noted and several formatting issues of the manuscript are changed. The Figures and the Tables are our own elaborations to describe the simulation models and the results of its application. For this reason the source of all these elements is not reported. Table 3 is an exception and the source of the data presented is reported in the last line of the table.

Authors need to edit the text. Sentences are too long and difficult to read. The text must be English proofread by a native speaking person. The file attached shows some of the issues found.

Reply to comment: the comment is noted and the manuscript is checked and revised accordingly the observation received.

Proper use of terminology is needed. For instance, we do not have shipping fares, but shipping rates or freight rates.

Reply to comment: the comment is noted and the manuscript is changed accordingly using the term of rates.

Also, the authors suggest a speed increase in some routes. Is this compatible with the current EU decarbonisation policy? please consider and reflect this in the outcome of the paper?

Reply to comment: the comment is noted and the manuscript is changed in the Conclusions considering the observation received.

Reviewer 2 Report

Despite of the goals you've set down for the literature review, still it is a goo practice to analyze sufficiently the theoretical background and adding one report is not the fulfillment of the request.

After corrections/adds made there are many text errors, i.e. missing spaces, redundant letters (before first key words) etc.

Author Response

Dear reviewer

thank you for your very useful suggestions. As follows you can find the response to any comment provided:

Despite of the goals you've set down for the literature review, still it is a goo practice to analyze sufficiently the theoretical background and adding one report is not the fulfillment of the request.

After corrections/adds made there are many text errors, i.e. missing spaces, redundant letters (before first key words) etc

Reply to comment: the comment is noted and the manuscript is checked and revised accordingly the observations received. About the literature review, three additional papers concerning SSS and modal split for freight are included in the review, mantaining anyway the goals of the literature review in the manuscript as already described in the first step of manuscript review process.